# Observational Evidence of the Need for Gender-Sensitive Approaches to Wildfires Locally and Globally: Case Study of 2018 Wildfire in Mati, Greece

**Anastasia Zabaniotou [1,2,*], Anastasia Pritsa [1,2] and E-A Kyriakou [1,2]**

[1]   Biomass Group, Chemical Engineering Department, Faculty of Engineering, Aristotle University of Thessaloniki (AUTh), 54124 Thessaloniki, Greece

[2]   Young Ambassadors of the Mediterranean (GAMe), Réseau Méditerranéen des Ecoles d'Ingénieurs, Ecole Centrale Marseille, UNESCO Chair in Innovation and Sustainable Development (651), CEDEX 20, 13451 Marseille, France

*   Correspondence: azampani@auth.gr; Tel.: +30-2310-996274 or +30-6945990604

**Abstract:** The study takes an equality justice perspective to compare resilience against the controlled management of wildfires, for an effective preparedness, which is a prerequisite for equitable mitigation. The objectives were (a) conceptualizing wildfire mitigation by exploring the ties with gender equality to wildfire hazards, (b) taking the case of wildfire 2018 in Mati, Greece, to contribute reducing the country's gender inequality, and (c) increasing resilience to climate change hazards by considering lessons learnt. The authors underscore the benefits of a workshop-based and instrumental case study methodology for unravelling evidence on the need for gender-sensitive approaches and tools for future planning at local, regional, and global scales. The case study unravels women's lack of preparedness to wildfires in Greece, their absence in decision-making for fire management, and the need for capacity building to transform communities' resilience. The literature research and the specific interviews conducted helped bring awareness to the wildfire's dynamics, in alignment with the fundamental aspect of gender equality, and to ground recommendations for socio-ecological resilience transition and gender-sensitive approaches in fire management, from reactive fire-fighting to proactive integration. Although in the geographical-context, the study can bring widespread geographical awareness, bringing insights for relevance to similar areas worldwide.

**Keywords:** climate change; wildfires; resilience; gender equality; TARGET project; Greece





## 1. Introduction

Earth's climate is changing. While our understanding of how climate change affects us is increasing, extreme weather is resulting in more and more natural disasters everywhere. Limiting global warming to 1.5 °C is related to ensuring a more sustainable ecosystem to remain below the relevant risk thresholds [1]. Failure to decrease global warming to 1.5 °C by 2030 could make climate change irreversible.

The IPCC report highlights the importance of limiting global warming to 1.5 °C, because climate change hazards are witnessed in every country causing disruption to national economies and affecting people's lives [2], with the latest Covid-19 pandemic disrupting the whole globe. Anthropogenic interventions in nature have created intense stresses to natural habitats and resources [3], setting the alarm for future natural hazards and disasters. Facing those hazards and limiting the global warming to 1.5 °C requires rapid, far-reaching, and inclusive changes in the socio-ecological systems and communities, towards increasing resilience at all levels [2]. To achieve reductions in risks from climate-change-related disasters would require societies to become more proactive, flexible, and resilient in integrated, multi-dimensional, and inclusive ways [4].

Climate change is a broad field and cuts across many sectors as well as institutions including national legislative bodies, local and regional governments, civil society organi-

zations, research, and academic institutions. It is a complex environmental and social issue, affecting a world that is characterized by deep-rooted unequal gender relations. Research shows that climate change affects women and men differently and when confronted with hazards, women and men have different needs, priorities, and possibilities, where the voice of women is not sufficiently heard and considered [5].

Gender equality and inclusion are of growing importance and focus in many sectors, including education, research, business, and governance for the 2030 Agenda for Sustainable Development that envisages building peaceful, resilient, equitable, and inclusive societies. SDG5 is a critical goal because its implementation can foster positive cascading effects on the achievement of all SDGs, which is directly connected to the nexus of education-sustainability. Socio-ecological resilience should integrate the fundamental aspect of gender equality. Women are the victims of climate change disasters. Gender equality is another parameter to be considered in climate-based resilience approaches. Vulnerabilities and gender inequality can impede the effectiveness and sustainability of climate change responses. Increasing gender equality has a positive impact on productivity, boosts problem-solving, and increases innovation, all of which are essential outcomes for tackling the challenges we face, from health to food security, from climate change to sustainable communities. Women's greater participation would not only be a social policy but would also enhance sustainability, resilience, and democracy. Policy planning should integrate natural and social capitals, ethics, and values, as agencies move towards the acceleration of the fundamental changes for a sustainable and resilient life. Therefore, to develop and maintain a sustainable and effective response to climate change and to increase resilience to climate change-based hazards, gender approaches must be an integral part of all policies and actions at all levels [5].

The climate-change-based challenges require everybody to negotiate the gender paradigm and fully integrate a diversity of actors. In light of the Agenda 2030, the fifth Sustainable Development Goal (SDG5) strives to achieve gender equality and empower all women and girls, at all levels and scales. Gendered analysis in climate-based disaster studies appeared in the late 1990s. Since then, there has been an increase in international literature dealing with the relationship between gender (women) and disaster, mostly recognizing the heavily male-dominated nature of formal disaster response and emergency services organizations [6].

However, despite a widespread commitment in SDGs in recent years, many fire organizations are slow to gender-sensitive approaches in doing business. With larger human populations and a climate-changing to drier levels, wildfires will continue to increase in the Mediterranean countries affecting ecosystem and community's vulnerability, leading to vegetation, soil, and human life loss [7]. In Greece, there is an absence of recognition in research, planning, and policy on important gendered issues surrounding wildfires.

*Scope and Objective of the Study*

The study contributes to the unravelling of the existing gender inequalities in wildfires management and coping. We wish to open a dialogue on the transition to inclusive socio-ecological resilience to wildfires, to spark awareness for changing stereotypes, empowering women's role in climate-change hazards and to share some recommendations based on lessons learnt from the Greek case study.

We wish to open a dialogue on the transition to inclusive socio-ecological resilience to wildfires, to spark awareness for changing stereotypes and empower women's role in climate-change hazards, and to share some recommendations based on lessons learnt from the Greek case study. We do not intend to give a broader spectrum of opinions from all relevant stakeholders and the local households' impacts, local business owners, local city officials and planners. This is because, as explained in this paper, the interviews with local stakeholders were not possible due to their displacement (the place was completely evacuated, and people were displaced from their homes in Athens; the place was a vacation and touristic place). Our intentions were to boost the case study with three top-personalities'

opinions on the situation in Greece, for tracing the vulnerabilities of the system to discuss the need for inclusive and gender-sensitive approaches at all levels, from the community to management bodies, from local to large-scale.

This study was presented at the Michelangelo2019 workshop. The Michelangelo2019 Workshop '*Natural Hazards Risks, Resilience and Gender Inequalities-Building bridges within the Mediterranean and TARGET project*' that took place in Rome, Italy, in June 2019, aimed at further awareness and perceptions of climate-based hazards and gender equality dimensions among students and faculty from the member institutions of the Network of Mediterranean Engineering Schools and Management (RMEI).

The study involved an interdisciplinary and complex thinking viewpoint of the subject. It had the objective to offer an understanding of wildfire dynamics through the lens of women's differentiated vulnerability but also in terms of alignment with the fundamental aspect of gender equality, as inspired by the TARGET project, which is a Horizon 2020 project, standing for "*Taking a Reflexive Approach to Gender Equality for Institutional Transformation*".

It recommends gender-sensitive approaches into wildfire management for resilience building for communities and individuals. By understanding women's vulnerability in wildfires, we can reorganize the preparedness and management of wildfires to include women-sensitive approaches in all strategies and at all levels. Women can play an important role in wildfires, making gender equality another parameter that must be considered in climate-based hazards.

It should be noted that the study of gender refers specifically to the "socially learned behaviour and expectations that distinguish masculinity and femininity" [6].

## 2. Research Methodology

The authors of this paper underscore the benefits of a workshop-based and instrumental case study methodology for recommending gender-sensitive approaches to increase resilience, cope with societal vulnerabilities and develop the tools for future planning at local, regional, and global scales.

By applying the backcasting methodology, which is a planning method that starts with defining a desirable future and then works backwards to identify policies and programs that will connect that specified future to the present [8], students in the dedicated Michelangelo 2019 Workshop identified wildfire climate change-based risks as a leverage point, derived actions towards a perceived ideal future and socio-ecological resilience as the suitable strategy to address it all. Backcasting is a scholarly and planning approach in fields related to sustainability, as an alternative to traditional planning approaches and a formal element of future strategic initiatives [9].

The study included an extensive and timely bibliographic search on the Internet, Google Scholar, Web of Science, Sage Publications Inc., Routledge, and ScienceDirect databases, and on international organizations reports, on wildfires hazards and gender equality.

It also included a case study that was presented by students in the Michelangelo 2019 workshop. We used the methodology of a case study because it is an empirical inquiry that investigates a contemporary phenomenon within its real-life context, especially when the boundaries between phenomenon and context are not explicit [10].

This case study could be characterized as an 'observational' cross-sectional study, conducted to gain insights into comprehension, recognition, and experience of gender equality role in wildfires in Greece, and to explore the resilience approach associated with this event. The case study we focused on concerns the wildfire of 2018 in the Mati village of the Attika region in Greece, aiming to unravel the links of gender equality and wildfires in general. It is an 'exploratory' case study that is used to explore situations in which the intervention being evaluated has no clear, single set of outcomes [11]. It can also be characterized as a 'descriptive' case study because it describes the phenomenon and the real-life context in which it occurred [11].

The case study can be also 'instrumental' and can be used to accomplish gender equality awareness rather than understanding, by providing insights into gender equality

and wildfires, for helping to refine an approach, while the case itself is of secondary interest. The case only plays a supportive role, facilitating our understanding of gender equality-sensitive approaches to climate-change disasters [12,13].

To boost the case study, three unstructured interviews/discussions with top-personalities in Greece were also conducted. Conducting interviews was chosen because this method of data collection gives voice to experts and lets them share their experiences and perceptions of situations, [9,14]. Interviews are the cornerstone of modern research used by both experienced and novice researchers [15], widely used because they allow in-depth analysis from a relatively small sample size [16]. By applying the small-scale sample size interview-based methodology, we identified wildfire climate change-based risks as a leverage point and took socio-ecological resilience as a suitable strategy to address it. We addressed the perceptions of the role of women in the time of wildfire 2018 in Attica and discussed the outcomes of 3 interviews with top-personalities.

The interviews of this study consisted of questions posed to conveners. To build the questionnaire, we first reviewed recommendations found in the literature on how to conduct interviews. The questions reflected on the wildfire conditions and preparedness, planning and management, relationships with gender and lessons learnt, posed in an unbiased manner. The interviews lasted approximately 45 min each, with key subjects conducted in the offices of the subjects.

### 2.1. The Michelangelo Workshop (MW)

This paper benefits from the Michelangelo2019 Workshop, which was designed to catalyze the development of perceptions and consciousness of students and scholars in Mediterranean Higher Engineering Education, by focusing upon the processes of learning on socio-ecological resilience and gender equality, to achieve a better understanding of our responsibility to help societies live sustainably, with equity and resilience to climate change impacts. Resilience and empowerment of women addressed in the Michelangelo2019 Workshop that was co-organized by students from the Mediterranean Engineering Higher Education, participating in the GAMe group (Young Ambassadors for the Mediterranean) (https://www.rmei.info/index.php/fr/programme/michel-angelo/what-is-game) belonged to the Network of Mediterranean Engineering Schools (RMEI) (http://www.rmei.info/index.php/en/).

The workshop was supported by the TARGET project 'Taking a Reflexive Approach to Gender Equality for Institutional Transformation' (http://www.gendertarget.eu/tag/rmei/), in which RMEI was a partner. TARGET served as the framework from which the research questions were derived.

The Michelangelo Workshop (MW) is organized every year. It is a neutral space where students become change-agents in addressing complex problems from a systemic perspective. Several volunteer-students co-organized the workshop2019 and carried out insightful presentations on gender equality approaches, in conjunction with sustainability and resilience to climate change-based disasters. Students also prepared related videos, interviews of key persons, narratives, etc. Some students identified wildfire climate change-based risks as a leverage point, and socio-ecological resilience as a suitable strategy to address it all, as it was the study of the wildfire of 2018, Mati, Greece.

Woman differentiated vulnerability during and after the wildfire, lack of women in decision making for fire management, and the very need for capacity building about this element to adequately transform the community resilience, were discussed.

### 2.2. The Network of the Mediterranean Network of Engineering Schools (RMEI)

The RMEI network (Mediterranean Network of Engineering Schools) was established in 1997, in France, with the mission of Sustainable Development. The network is affiliated to UNESCO UniTwin chair of Sustainable Development innovations. Its vision is Sustainable Development and Peace in the Mediterranean region through education on sustainability,

responsible research, and innovation in engineering and inclusivity. RMEI today involves 28 members.

The network is a member of the consortium of the EU HORIZON2020 TARGET project that aims at initiating sustainable institutional change in seven institutions in the Mediterranean region, including the RMEI network. The participation of RMEI in the TARGET has an impact on the collective transformative agency (professors, students, leaders) who are creating a Community of Practice in gender equality.

Figure 1 depicts the relation of this study with the RMEI, GAMe, Michelangelo workshop (MW), and Target project.

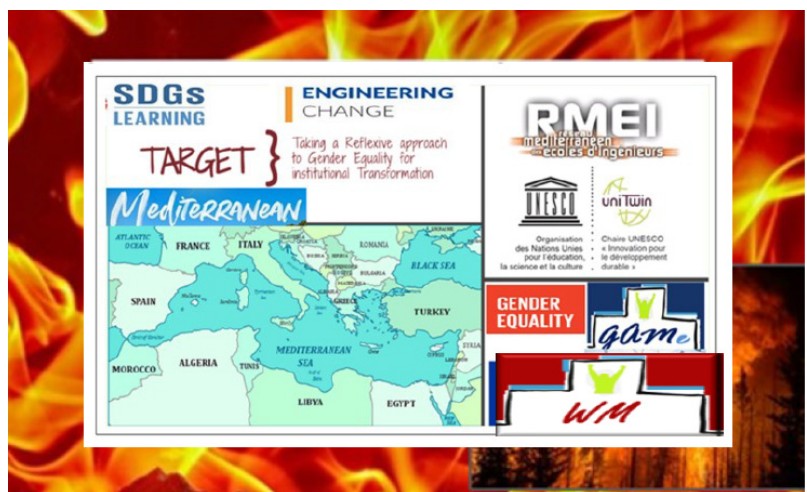

**Figure 1.** The interconnections of the study with RMEI, GAMe, Michelangelo2019 workshop, and the TARGET project.

## 3. Research Questions and Conceptual Frameworks

The following chapter presents the bibliographic search and addresses the following research questions:

- Is there any link between wildfires and genders?
- Why address gender issues in wildfire management and policies at all levels—is it important?
- What action should we take?
- Why does gender matters in resilience programming?

The following frameworks are shortly defined to address those questions.

### 3.1. Natural Disasters

The term "natural disasters" refers to all catastrophic phenomena such as wildfires, floods, earthquakes, volcanic eruptions, tsunamis, tidal waves, storms, that can be fatal for human communities [17]. These hazards involve life and property loss, social and economic disruption, infrastructure and natural resource damage, air quality deterioration and accumulation of any post-disaster wastes [17]. They are complex phenomena of social, ecological, economic, gender and political dimensions, interconnected through wicked loops [18].

### 3.2. Wildfires

Fire is an earth's system process that depends on the local vegetation characteristics, climate, and human activities [19]. It has impacts on society, the economy, and the ecosystem [20]. Wildfires are a global phenomenon of anthropogenic interventions on climate change. Global heating increases the intensity of wildfires, having an impact on the achievement of the Sustainable Development Goals (SDGs) [21].

Climate change leads to a long-lasting and dry summer of high temperature [22]. It results in the abandonment of cultivated landscapes, and to biomass accumulation that can fuel fires [23]. Some summer human activities that involve campfires and barbecues can be sources of fire [24]. The feedback of this phenomenon is the impact on the biogeochemical cycles, partial or complete loss of vegetation, and the disintegration of physical properties of the soil [25]. Other results are burnt houses, infrastructure and landscapes, damaged local ecosystems, disrupted local and national economies and societies [21]. Wildfires can cause loss of human lives and properties, displacement, stress to people who have to flee them, and air-pollution due to the release of harmful pollutants. These include particulate matter and toxic gases that are released into the atmosphere and increased post-hazard wastes, disturbing the whole socio-ecological system and community. Wildfires also affect the $CO_2$ cycle, as depicted in Figure 2.

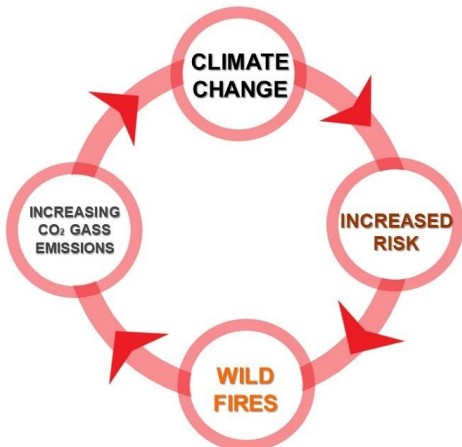

**Figure 2.** Climate change-based wildfire and $CO_2$ cycle.

Wildfires associated with loss of human life and property are becoming increasingly common in the Mediterranean and globally, highlighting that the current management approaches are not sufficient and that new social-ecological inclusive approaches are needed towards increasing resilience of human and ecological systems [4].

### 3.3. Vulnerability

A human or ecosystem's vulnerability is defined as the state of susceptibility to harm from the exposure to stress and the absence of capacity to adapt [26]. It is a function of exposure, sensitivity, and adaptive capacity [27]. It is a multidimensional process affected by ecological, social, political, and economic forces, interacting from local to international scales [28,29].

The decreased regeneration ability requires substantially longer periods of ecosystem recovery before reaching the pre-fire tree density, [29]. Water shortage, which is the case in Mediterranean countries, is another significant parameter inhibiting the regeneration of trees and creating ecological vulnerability [30].

### 3.4. Adaptability-Transformability

Adaptive capacity is defined as the system's robustness to change [31]. It is the ability to deal with adversity (coping) and includes strategies used by individuals to adapt to adverse or stressful circumstances. Adaptability is also the capacity of a social–ecological system to adjust its responses to external drivers and internal processes. Adaptability is part of resilience [32].

Integration of adaptation and mitigation practices requires comprehensive approaches, close cooperation, synergy and coordination among policy planners, institutions, local communities, and the global society [33] (Figure 3).

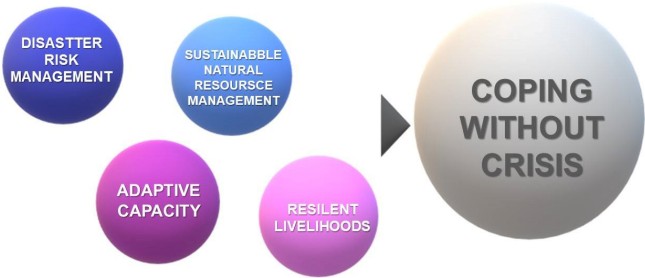

**Figure 3.** Factors of coping and adapting.

*3.5. Mitigation*

Natural hazards are complex events whose mitigation generated a diverse field of specialized expertise. In the application of risk mitigation for a wildfire, the challenges are multi-dimensional, related to current knowledge, R&D, funding, institutional priorities, literacy, and intellectual property. For predicting risks and managing hazards it requires understanding the risk of the area, the environmental and geographical knowledge regarding vegetation, topography, land use, and population distribution [5].

Risk mitigation shapes the intersecting public policy discourses of mutual responsibility and policy [34].

*3.6. Transformability-Resilience*

Transformability and resilience are central concepts in addressing the dynamics of complex social–ecological systems, across multiple scales. Transformability is the capacity to create new stability domains for development, into a new development trajectory and to cross thresholds into new development trajectories that come from a transformational change at smaller scales, enabling resilience at larger scales [35].

Resilience has various definitions in various domains:

❖ Engineering resilience is the degree to which a system approaches steady-state and returns to equilibrium after a disturbance [36].

❖ Resilience to natural disasters is the system's ability to absorb disruption to preserve its previous function and construct [36]. It is measured by the scaled change happening to the ecosystem, as the ability for the ecosystem's self-organization and community to learn from it and adapt [37].

❖ Resilience in ecological systems is the amount of disturbance that a system can absorb, without changing the stability domains [38]. The loss of resilience brings a change in the system state, signaled as a resource crisis [30]. Ecological systems are coupled with social ones, since humans are part of the ecosystem [39].

❖ Social–ecological resilience is "the capacity of social–ecological systems to absorb recurrent disturbances to retain essential structures, processes and feedbacks" [40].

❖ Equitable resilience identifies critical issues for engaging with equity in resilience practice [40]. Taking a gender-responsive approach to mainstream and accelerated climate actions across the globe, towards empowering women. Women and girls should be the central drivers of these actions as they provide safety and security for their families, and educate future generations [32].

Learning, trust, and engagement are components of social resilience. Social learning can be facilitated by the recognition of uncertainties, monitoring and evaluation by stakeholders [38]. The resilience approach needs technology transfer, education, training, scientific cooperation, participation, equity, inclusion, and gender-balanced leadership. Safety of firefighters, ecosystems, communities, and individuals should be of priority [41]. The development of holistic resilience needs to include broad features of human rights and livelihoods, social and gender vulnerabilities, and other adversities such as poverty, food

security, clean water, good health, education, and participation in the economic life of the country complying with the SDGs agenda [42].

### 3.7. Control-Based Wildfire Management

Fire management is the management of land by safeguarding life, property, and resources through prevention, detection, control, restriction, and suppression of wildfire in forests and other rural areas, by incorporating fire policies and actions, involving the strategic integration of several factors, to support the development of Fire Management Plans [42]. These factors require specialized knowledge of:

- fire regimes,
- probable fire effects,
- level of risk,
- level of forest protection,
- cost of fire-related activities,
- appropriate technologies required.

When a system has shifted into an undesirable stability domain through irreversible changes, the current management alternatives address one of the following or all [30]:

1. Restoration of the system to a desirable level.
2. Allowing the system to return to a stable domain by itself.
3. Adapting to the system that has changed.

Many organizations, scientists and policymakers adopt strategies that they differ from in rationale:

a. The control-based management approach.
b. The resilience approach.

The controlling rationale has its roots in engineering and economics [43]. Usually, current fire management activities are activities of a fire control approach, such as [44]:

- Prevention.
- Early warning.
- Detection.
- Mobilization.
- Appropriate use of natural or human forces.
- Reducing the accumulation of residues and wastes from commercial or non-commercial activities.
- Rehabilitation of ecosystems damaged.

### 3.8. Resilience-Based Wildfire Management

A key issue for successful fire management under climate change is the adaptive capacity of the system, which depends not only on the available scientific and technical knowledge, but also on the social, economic, and political components associated with the implementation of the different adaptation options [45].

The resilience-based rationale has its roots in ecology and transformation [35,46]. In the future, a resilience approach for the climate-change based disasters management and planning that takes ecology in deep consideration should be addressed [46]. As the wildfire risk is a complex socio-ecological issue of relationships with fire and wildfire losses are expected to increase, it is critical to have a diversity of representation in fire management and to recognize the value that this diversity can bring to the discipline [47–49]. However, despite the extensive fire research, knowledge of adaptation strategies, and resilience is limited. The barriers to effective implementation of the holistic resilience plans might be the high costs, dependence on political decisions, knowledge, landowners, and stakeholders' interests.

Many strategies for resilience address a variety of purposes aiming at increasing the buffering capacity of the system, managing processes at multiple scales, and nurturing the sources of renewal [31].

These strategies include:

1. Restoring or maintaining ecological resilience.
2. Increasing individuals and community's resilience.
3. Reducing the resulting loss of life and properties by wildfires, and calls for the involvement of socially diverse local communities, as because people from different cultural backgrounds respond differently to wildfire risk, as do men and women [50].

It is, therefore, necessary to know the impacts of regional climate change on the human–nature ecosystem to be able to develop adaption plans and policies that consider all biome changes in the area, as schematically depicted in Figure 4.

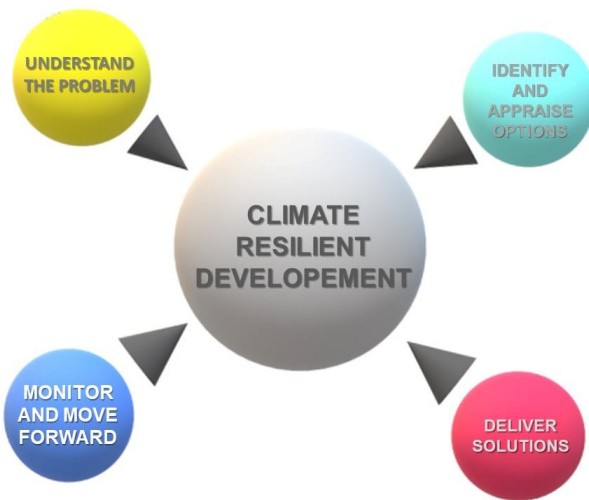

**Figure 4.** Process for climate resilience development.

The factors and facets of resilience are many. Resilience is ecological and social. It has a spatial character, and it is impacted by environmental characteristics, ecosystem attributes and processes, disturbance, landscape composition, and configuration, as depicted in Figure 5.

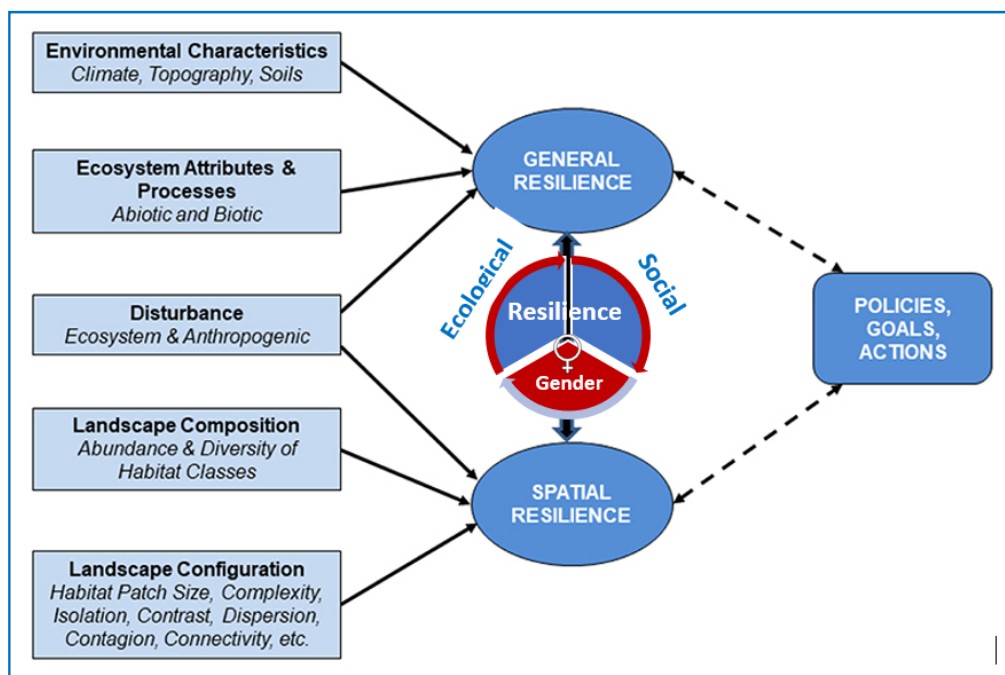

**Figure 5.** Factors and facets of resilience.

### 3.9. Gender Equality and Climate-Based Hazards

Ecological degradation creates economic disruption, putting women and girls at increasing risk. While climate change is demonstratively unique, unprecedented impacts on women and girls everywhere are witnessed. Gender-responsive actions are needed to address these disruptions, while also strengthening gender equality [51].

Many efforts are underway to mitigate climate change impacts on socio-ecological systems. Resilience is acknowledged both explicitly and implicitly in the proposed SDGs targets. However, although, all 17 SDGs set by the United Nations aim at sustainable development that is coupled with the fulfilment of social and psychological human needs for health, prosperity, and unity, systemic resilience requires a more holistic approach to the SDGs, realizing human rights and social equality [52]. Resilience means that the whole system or part of it after the disturbance comes back to its previous equilibrium state or it goes to a new state [53]. The UN with the Gender Action Plan seeks to advance women's full, equal, and meaningful participation and promote gender-responsive climate policy and the mainstreaming of a gender perspective (https://unfccc.int/topics/gender/workstreams/the-gender-action-plan).

People need not only learn about hazards, prevention, reduction, and avoidance in the future but should also be resilient [54]. During and after hazards, people and environment need to be strong enough to adapt to the new reality. Hazards emergency planning and mitigation strategies must be tailored to the population affected by the hazard, with regards to socially vulnerable groups, women, and multiethnic and multilanguage populations [55]. Researchers argued that language barriers prevent people who do not speak the national language, from receiving emergency information, in countries with multilingual populations [56]. The need for analysis of gender concerning class, ethnicity, age, disability, religion, sexuality, parenthood, to understand how these intersections contribute to and reinforce inequalities are highlighted in a book edited by Susan Buckingham and Virginie Le Masson (eds.) (2017), [57].

Adopting a broader approach to resilience thinking allows managers, stakeholders, and policymakers to consider adaptive and transformative responses that can change conditions of the ecosystems (fuel conditions, vegetation) and social systems (institutions) [58]. Policies must account for power dynamics and gender relations to improve their effectiveness and to avoid exacerbating gender inequalities [57].

Disasters have several complex and subtle ways in gendered consequences. Women tend to be more at-risk during disasters like megafires. Although studies have shown that disaster fatality rates are much higher for women than for men, due in large part to gendered differences in the capacity to cope with such events [59], efforts to promote social–ecological equitable resilience to wildfires are falling back. Especially, the culturally and historically distinct gender relations that underpin wildfire resilience are falling short, are insufficiently funded, and lack broad public support. [60].

Gender vulnerability associated questions are elements of social science research. Disaster studies that adopt the use of gendered analysis appeared only in the 1990s, by Australian and American authors who argued that there are substantial gendered differences in disaster preparation and response [61,62]. Meagan Tyler, 2013 advocated that the social construction of masculinity-femininity must be considered in the development of education programs, in order to become more resilient in future [62]. In 2014, Eriksen C. (2014) examined outreach initiatives in Southeast Australia, specifically targeting the awareness and preparedness of women [63]. In 2016, Eriksen et al. (2016) considered the issue of gender difference in mentoring, training, and leadership [64]. In 2018, Reimer and Eriksen [65] unraveled the links of gender, leadership, and wildland fire culture, by providing clear insights into gender discrimination into leaderships, and highlighted the trade-off between gender diversity and excellence among research participants. In 2018, in her book 'Gender and Wildfire Landscapes of Uncertainty', Christine Eriksen examined wildfire awareness and preparedness amongst women, men, households, communities, and agencies, at the interface between the city and beyond. She did so through an examination

of two regions where wildfires are common and there is a major political issue—Southeast Australia and the West Coast of the United States. Culturally and historically distinct gender relations underpin wildfire resilience [60].

Virginie Le Masson (2017) leading the research theme 'Gender and social equality' within the BRACED program, in her working paper, presents a synthesis of four case studies documenting strategies towards building gender equality through resilience projects, by drawing on the experience of non-governmental organizations (NGOs) involved in the implementation of the Building Resilience and Adaptation to Climate Extremes and Disasters (BRACED) projects. She also provides a set of recommendations to point out areas where further research is required to increase the understanding of resilience to climate extremes and long-term changes, and to suggest how donors and funding can best support efforts to build resilience [66].

The book 'Understanding climate change through gender relations' edited by Susan Buckingham and Virginie Le Masson (eds.), broadly documents gender relation in the context of climate change [13]. Gender discrimination is clear within almost all fire wildland firefighting institutions and fire management bodies in the Mediterranean countries, which are strongly hierarchical and have a masculine culture [64].

Gender vulnerability is a social factor measuring climate-related impacts [67]. Gender refers to dynamic social processes that are deeply rooted within society that lead to socially constructed gender roles, identity, gender norms, and behavior that is considered appropriate for men and women. The reasons are:

■ Belief that femininity equals weaknesses.
■ Preconception that women are not linked to leadership excellence.
■ The dominating situation of built-in defensive mechanisms that make cultural change difficult.

Addressing gender discrimination requires cultural change. Fostering gender diversity requires transforming leadership practice to include open engagement with gendered cultural norms. Institutions should focus on the learning and understanding of cross-scale interactions of the social-ecological systems. Vulnerabilities must be brought to the forefront of developmental policies and practices [26]. Disaster researchers and sociologists are focused on social, economic, political, and cultural factors that determine different levels of vulnerability to natural hazards, between nations, societies, and individual. Natural disasters and gender inequalities are socially rooted in cultural, political, and socio-economic conditions, varying between societies that have complex consequences for women, men, and others.

*3.10. Towards SDGs Agenda 2030*

Climate change and global heating have increased the likelihood and intensity of wildfires, which have an impact on the achievement of the Sustainable Development Goals (SDG). SDG5 concerns gender equality, although women play a critical role in all SDGs. Gender equality and empowerment is recognized as the objective but also part of the solution. At the international level, the interlinkages between the 2030 Agenda and the commitments for gender equality connected to the implementation of the Paris Agreement offer an opportunity for countries to coordinate their actions and promote gender, climate action, and social progress, at the national level [31,32].

**4. Case Study**

*4.1. Wildfires in the Mediterranean*

Wildfire fatalities remain a significant problem in Mediterranean Europe [68]. Many catastrophic cases were recorded in recent years in Southern Europe. Some of these were remarkable forest fire events [68]:

❖ Mati (Greece, 23 July 2018, 99 fatalities)
❖ Central Region (Portugal, 15 October 2017, 53 fatalities).
❖ Pedrogao Grande (Portugal, 17 June 2017, 66 fatalities).

- ❖ Horta de Sant Joan (Tarragona, Spain, 21 July 2009, 5 fatalities).
- ❖ Makistos–Artemida (Peloponnese, Greece, 24 June 2007, 30 fatalities).
- ❖ Riba de Saelices (Guadalajara, Spain, 17 July 2005, 11 fatalities).
- ❖ Ikaria (Ikaria island, Greece, 30 July 1993, 13 fatalities)
- ❖ Curraggia (Sardinia, Italy, 27–28 July 1983, 9 fatalities).
- ❖ Agueda (Portugal, 14 June 1986, 16 fatalities).
- ❖ Armamar (Portugal, 8 September 1985, 14 fatalities).
- ❖ La Gomera (Canary Islands, Spain, 11 September 1984, 20 fatalities).
- ❖ Lloret de Mar (Girona, Spain, 7 August 1979, 21 fatalities).
- ❖ Sintra Mountains (Portugal, 7 September 1966, 25 fatalities).

In recent decades, not only the number of fires increased, but also the areas that were affected [69]. In the European Union (EU), most wildfires (up to 90%) occur in Mediterranean countries [70]. Social and environmental concerns are raised due to the frequency and intensity of wildfires, while resilience faces uncertainty [19]. Areas quite prone to fire are those with Mediterranean climate because they have a dry and warm climate, coupled with easily flammable vegetation and human activities that have created unsustainable interventions to the ecosystem [71]. Specifically, the countries of Southern Europe (i.e., Italy, France, Spain, Portugal, Greece) are facing significant impact from wildfires [72]. For all Southern European countries, wildfires exceeded 1000 hectares in size, during the period of 1989–1993, with a rate of ~0.1% of a total number of fires, which was responsible for ~27% of the total area burned [73]. Comparing the number of wildfires with the corresponding area burned, it became clear that even a small number of fires can destroy an area [74].

Although wildfires in Greece do not represent a large part of the total number of wildfires concerning the rest of Southern Europe, it still has a disproportionally high number of catastrophic wildfires. During the period of 1986–1995 large fires (greater than 1000 ha) representing approximately 6% of the total fires, 2/3 (66%) of the total burned area were destroyed [74]. Over 50% of the total burned area (approximately 162,000 ha) was the result of only seven large fires that occurred during summer 2000 [75].

Before the catastrophic wildfire of 2018 in the Attica area (Mati village), near Athens, other wildfires were recorded in the same region. On 21 July 1995, a large wildfire broke out in a pine forest on the Mountain of Penteli in Attica, which burned about 251 km² of land. Approximately, 105 buildings were heavily damaged or destroyed, with, fortunately, zero loss of human lives. On 2 August 1998, a wildfire occurred northeast of Athens, re-incinerating any regenerated vegetation [76]. Hundreds of properties and public buildings were destroyed or seriously damaged and human lives were lost. On the 28 July 2005, another wildfire resulted in that burned land and many houses [77]. On the 21 of August 2009, a wildfire took place in Eastern Attica, causing a total of 850 km² of burned land, which mainly constituted pine tree forest, and 72 houses, while several local communities were heavily affected [78].

The effect of wildfire on local vegetation is high. In the post-wildfire period, tree regeneration is slow because the climate has become warmer and drier [30].

### 4.2. The Wildfire of 2018 in the Mati Village, Attica Region, Greece

Mati is a village located on the east coast of the Attica region, 29 km east of Athens. It is a popular holiday resort and tourist destination. The harbor serves as an access point to the Aegean Sea (Figures 6 and 7).

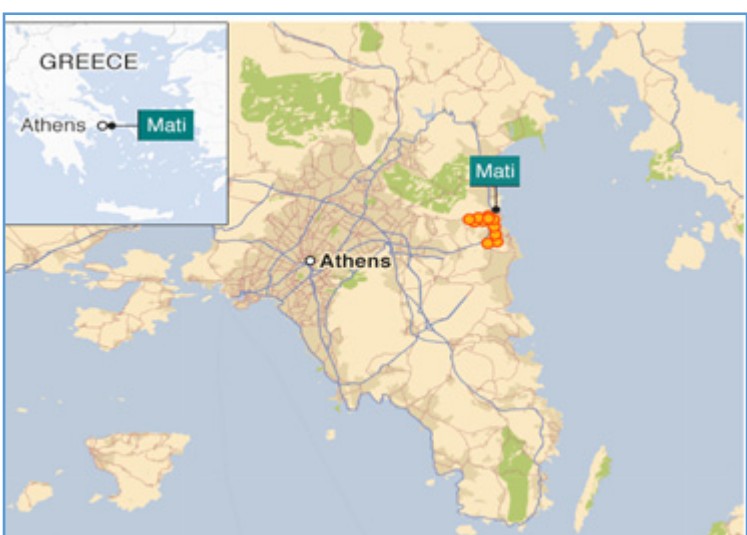

**Figure 6.** Location of Mati, Attica, and its vicinity with Athens (Source NASA).

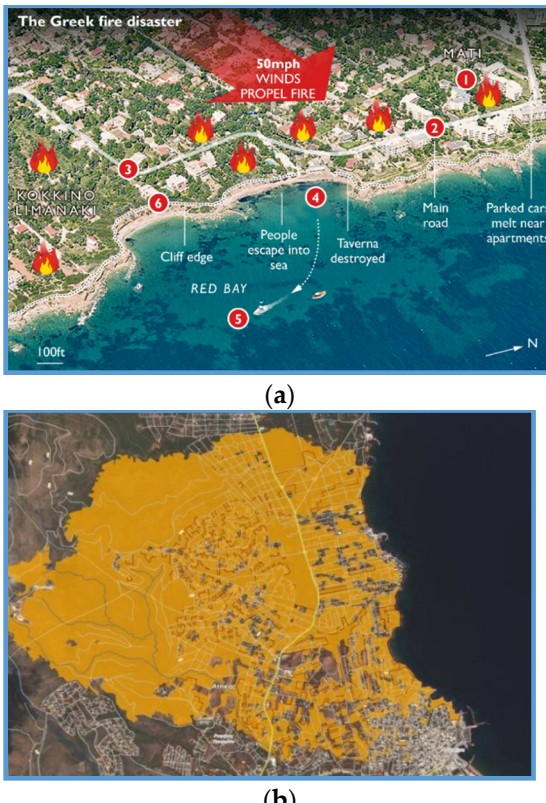

(**a**)

(**b**)

**Figure 7.** (**a**) Location of the wildfire, Mati, Attica, Greece (Source: Emergency Live); and (**b**) devastation of residential areas (Source: EU satellite service Copernicus).

The wildfire on the 23 July 2018 in Mati is referred to as the deadliest natural disaster in the history of the Modern Greek state, according to EFFIS (https://effis.jrc.ec.europa.eu/).
The devastating direct results were [78]:

- 99 humans lost their lives (55 women, 45 men, including children)
- 164 people were heavy injured.
- Over 700 residents were evacuated or rescued.
- More than 4000 residents were affected.

- Hundreds of square kilometers of pine tree forest were burned.
- More than 1500 buildings and houses were destroyed/damaged.
- 305 vehicles were burnt.
- Severe damages were caused to electricity, telecommunication, and water supply network.
- Besides the direct losses, long-term impacts involved the following [77].
- Health issues from impaired air quality, due to wildfire smoke and heavy metals and plastics melting.
- Disruption in tourism, business, and recreation revenue.
- High accumulation of different kinds of post-hazard wastes.

*4.3. Interviews for Boosting the Case Study*

Three interviews were conducted with three key-people (two men and one woman), to approach the issue from different angles:

The first interview was conducted with the Fire Brigade Operations Coordinator (male). The second interview was conducted with a journalist/meteorologist (male) on Greek TV. The third interview was conducted with a University professor (female).

Then, we analyzed these interviews. Based on the results, we provided a detailed checklist aimed at resilience dialogue. The replies were compiled in the discussion of this study, which were focused on presenting arguments for the enrichment of action research methodology on differentiated vulnerabilities, adding arguments and crucial inspirations for future planning-approaches.

All reviewers argued that the reasons for the fires catastrophic impacts were weather conditions, a lack of urban planning, type of topography, type of dense vegetation (pine trees), and the following:

- Lack of awareness and preparedness.
- Lack of evacuating skills by the residents.
- Lack of trust in the woman.

The three respondents of this interview-based study argued that for coping with the extreme wildfire phenomena, a policy shift to prevention, preparedness, and resilience development is required. The increase of wildfires hazards in the 21st century requires investments in wildfire personnel and infrastructure, efforts for community participation and women inclusion, training, organization of a capacity building, greater wildfire awareness, and improved planning to reduce the impact of fire, as much as possible. The three respondents highlighted the women's vulnerability to wildfire, as witnessed in the wildfire 2018 in Attica, Greece. Patriarchal dominated fire management structures, lack of trust in women's capacities at the time of the disaster, led to more life losses.

The specific comments by the interviewed person are presented below:

4.3.1. Interview I: Preparedness to Wildfires in Greece

The 1st interview was related to the management plans and preparedness to the wildfires, conducted with the coordinator of the Fire Brigade Operations Coordinator (male). He argued that there was not sufficient public awareness and preparedness in the country, related to forest or other fires, as compared to the level of preparedness for earthquakes. The Greek population does not know how to react in the case of wildfire, due to the less effort attributed by policymakers and regional administrators to increase awareness and preparedness to wildfires.

He concluded that there is a need for awareness and actions through education, media, and other initiatives. It was also confirmed that the Fire Brigade Operations were traditionally dominated by men. Women were absent in the body of firefighters and in the leadership of fire management.

### 4.3.2. Interview II: Global Warming and Wildfire Hazards Links

The 2nd interview was related to questions on the interrelation of climate change and global warming with wildfires. It was conducted with the state-run weather channel presenter on TV (male). He confirmed that anthropogenic $CO_2$ emissions directly contributed to the extreme temperatures documented in the Mediterranean and globally, in recent years. The extreme weather conditions result in natural disasters like wildfires. He argued that actions to tackle climate change are of high importance, along with changing human behavior towards the environment, ecosystems, and equity. He also mentioned that there is a need for awareness, preparedness, and gender-equality-sensitive approaches.

### 4.3.3. Interview III: Gender Equality and Wildfire Management

The 3rd interview brought insights on gender equality links to climate-based disasters and addressed the role of women during the wildfire of 2018. During this tragedy, women faced the greatest risk as they got in action to save not only their own lives but also those of their children and elderly people of the family. According to the Greek style of living, mothers, children, and grandparents mainly consist the population of vacation places in summer during weekdays, as is the case in the Mati village, while men are occupied during the week with their work in the city. Women who were present at the time of the wildfire eruption were primarily not engaged in productive activities like income generation through wage labor and entrepreneurship. They were responsible for childbearing, and household maintenance, including cooking and caring for elderly family members.

Additionally, cultural norms related to gender roles limited their ability to respond to or make quick decisions in the face of a fire event. Women were not listened to during the panic when they proposed pathways they knew for rescue. Access to the beach was difficult due to the steep slopes of the coast in the eastern part of the area. Consequently, people had difficulties approaching the coastline, to reach a safer environment. Some women knew the way. Unfortunately, in the panic, nobody was listening to them, and were waiting to be guided by men. This had a more tragic end, resulting in loss of more lives.

The male-dominated world of wildland fire is based on pre-existing cultural perceptions of masculinity. This is a matter of concern, as it seems that the society does not trust or rely on women, when in panic and during a crisis of natural disasters. People still inherently do not understand the knowledge and skills of women, not equating their leadership to that of men.

Regarding gender differences concerning disasters, studies showed that women present a significantly higher frequency of every type of mental health symptom, after a disaster. Women seem to have limited access to emergency response sources, and they have a degraded role in programs and structures of disaster management, which makes them more vulnerable, and their suffering in disasters is disproportional in a post-disaster time [79,80]. Although the role of women in the post-hazard time, as nurses, doctors, veterinarians, volunteers for the cleaning up of the area and organizing meals is important, the lack of women in decision-making and fire management positions of the related institutions and the leadership of the Fire Brigade is still lacking behind in Greece.

## 5. Discussion and Lessons Learnt

Crises can be windows of opportunities to bring social–ecological transitions from one state to another, recombining source of experience, knowledge, and innovation [31]. Women are at the center of the climate change challenge, as recognized by the key gender aspects of climate change actions identified by the Global 2030 Agenda, the United Nations Framework Convention on Climate Change (UNFCCC), and climate finance mechanisms like the Global Environment Facility (GEF) and the Green Climate Fund (GCF) [74]. Discussing resilience as a transformational change at smaller scales might enable it at larger scales [81,82]. Discussing a transformative change needs to consider the system's dynamics and the concepts of gender vulnerability, adaptation, and mitigation.

Gender equality is a wicked problem, interconnected with overlapping complexities inherent in our society, which create symbiotic relationships that connect issues. Therefore, gender equality is needed for coping with the global challenges of climate change, health, security of food, water, waste, energy, and biodiversity. Gender inequality can impede the effectiveness and sustainability of climate-change responses. Increasing gender equality has a positive impact on productivity, boosts problem-solving and increases innovation, and creates sustainable and inclusive communities. Women's greater participation and empowerment would enhance sustainability, resilience, and democracy. Policy planning should integrate natural and social capitals, ethics, and values, as agencies towards the acceleration of the fundamental changes for a sustainable and resilient life.

The shock of the wildfire of 2018 in Mati, Greece raised from a set of sources and anthropogenic maladjustments of the socio-ecological system, provides us with an opportunity to discuss adaptability and resilience and open a discourse on gender equality with natural hazards, a topic that was never discussed in the country and is completely missing from the research and policy.

The picture of the role of women and how women are affected by the wildfire in Mati, Greece, is complex. The fact that the cultural context in Greece during high-fire seasons suggest women stay at home to care for elderly and kids, and perhaps experience the brunt of evacuation efforts, already suggests that women take an active role in evacuation decision. However, it is also evident that women need empowerment for Greek communities to fare better during natural disasters. For an effective disaster risk management implementation of both proactive and reactive strategies before, during, and after the event should be considered. Although women can play a leadership role in early warning systems and wildfire management and evacuation, they are missing from all related agencies in Greece. Perhaps, lessons learnt from that experience is something that can help with the pre-event disaster risk management decision. The engagement does not necessarily have to be masculine, per se, like firefighters, but rather can be strategic and at the policy-making level.

The wildfire of 2018 in Attica made evident that there are gender differences in the impacts of climate change, responses to climate change, due to the different roles and responsibilities of women and men, which vary by the socioeconomic level and local inherent cultural norms. It was made evident that while women's actions are part of responses to disaster events, they are excluded in Greece from official emergency response agencies, due to the heavily male-dominated nature of formal disaster response and emergency services organizations. Compared to men, women face challenges in accessing all levels of policy and decision-making processes. These results in women being less able to influence policies, programs, and decisions that impact their lives, their family, and the community.

Some more subtle bias exists in Greece concerning wildfires and disasters managers that have gendered consequences. These biases concern the perception of people that wildfire management is a masculine world. In the Mati 2018 wildfire more women died, which is not the result of some biological differences between men and women, but the result of socio-political factors, including gender inequality (social and economic positions in society) and care-giving responsibilities for children and the elderly, which often impede a woman's ability to escape imminent danger.

Public data on male and female fatalities show that from 99 lost lives 55 were women, and 45 men (including children) while 164 people were heavily injured. These data show an overrepresentation among the victims (deaths), while the number of publicly provided injured people (in news, newspapers, etc.) does not distinguish between men and women, which shows the lack of gender sensitivity and importance on the accounting of injuries in Greece. A previously published research concerning wildfires in Greece that examined a database of 208 fatalities that occurred in 78 forest fires in Greece between 1977 and 2013, showed that male and older individuals' were overrepresented among the victims [82]. We argue that this does not depict a real situation of male and female fatalities of the civilians in wildfires in Greece, because the above-mentioned study included the deaths of firefighters,

forest service officials, and aircraft crews, which does not depict women's fatalities because these agencies are male-dominated in Greece and these data provide a different picture of gender in civilian fatalities. Additionally, this shows that gender-sensitive approaches are even missing in calculating the fatalities in Greece.

There were also substantial gendered differences in wildfire preparations, as evidenced by the interveners.

It might be criticized that the number of interviews was small, but we argue that this was cross-sectional and that the interviews with individuals affected by the wildfire were difficult, as the trauma was recent and due to their displacement in Athens. For this reason, in this study, we did not conduct interviews with a large number of participants (victims and inhabitants of the place). Although this would have also demonstrated emerging evidence of psychological problems associated with the event, we were limited to experts and three top-personalities in the country from different sectors and levels of experience, related to climate change, wildfire management, and gender equality approaches in academia, education, research and leadership, in order to explore how they perceive the questions:

'*Have the risks to wildfire hazards and resilience links to gender equality*?'

Indeed, the number of interviews performed were few, but the interviews were cross-sectional. We did not intend to do a survey with civilians' opinions. Interviews with individuals/civilians were not conducted, due to the displacement of the civilians affected by the wildfire. Interviews with inhabitants would have also demonstrated emerging evidence of psychological problems associated with the event, but it was not possible to meet those people because the trauma was recent. In this study, we did not conduct interviews with many participants (victims and inhabitants of the place) because it was a sensitive-case. We were limited to 3 top-personalities in the country from different sectors and levels of experience related to climate change, wildfires management, and gender equality approaches in academia, education, research, and leadership. We wished to explore how these persons perceived the question: '*Have the risks to wildfire hazards and resilience links to gender equality?*', because, in Greece, gender dimensions remain out of the central theme of climate hazards and management, although historically Greek women displayed enormous strength and tend to be very effective at mobilizing communities in the event of disasters. Often, they had a clear understanding of what strategies are appropriate at the local level. Historically, it was proven that if women have the capacity throughout the entire disaster cycle (preparing for hazards, managing after a disaster, and rebuilding damaged livelihoods), they can increase the community's resilience.

Australia is a more advanced nation in gender-sensitive approaches to the planning and management of wildfire. Australian authors also suggested that a gender-sensitive analysis of climate disasters needs to go beyond understanding 'gendered vulnerabilities' and to examine how the socially constructed societal expectations of women and men that underpin traditional views of climate disaster management as 'men's business' persist today [83]. Gendered dimensions exist in the following stages of disaster [84]:
Risk exposure.

1.  Perception of risk.
2.  Preparedness behavior.
3.  Warning communication and response.
4.  Physical impacts.
5.  Psychological impacts.
6.  Emergency response.
7.  Recovery.
8.  Reconstruction.

Taking into considerations the Australian research and adding the insights of the present study, we could advocate that women's greater participation would not only be a social policy, but would also enhance the sustainability and effectiveness of climate change responses. Mainstreaming gender in climate change policies and programs can minimize

these outcomes. Planning and execution cycle of climate change policies and projects must include women's needs and contributions.

Therefore, we co-advocate the following:

- Ensuring the equal participation of men and women in implementation, adaptation, and mitigation to climate changes disasters.
- Ensuring women can act as agents of change at different levels of adaptation and the mitigation process.
- Promoting mitigation approaches that are aware of gendered implications.
- Developing equal participation in the deployment of financial resources, particularly at the local level.
- Taking a gender-sensitive approach to creating, developing, and strengthening institutional, systemic, and human-resource capacity-building, to foster gender balance in decision-making, and accessing means and tools for the implementation of mitigation or adaptation actions.

## 6. Recommendations

Women should not only be victims of climate change, but should also be active agents of change towards creating a greater level of resilience. The use of the commitment to climate change and related fundraising can go together with transformative resilience, to have a positive impact on achieving women's equality to perform certain types of masculinity, in order to achieve social acceptance or personal self-worth.

While the focus of this research is wildfires, we wish to further provide some recommendations from lessons on gender equality from other hazards (e.g., floods, hurricanes, tornadoes, earthquakes) that have common features; these are all acute onset events and likely have similar management approaches.

### 6.1. Resilience Planning

For developing more resilient communities, planning is needed based on the lessons learnt from this place-based case study and from global experiences. Actors contributing to wildfire vulnerability are both social and ecological and they need strategies for resilience planning with consideration of some principles, as depicted in Table 1.

**Table 1.** Recommendations for strategies and resilience planning principles.

| No | Principle | Measures |
|---|---|---|
| 1 | Estimate the exposure vulnerability [84]. | - Vulnerability exposure needs estimation.<br>- Fuel conditions in the area must be known and well estimated.<br>- Less flammable building materials used in construction in the area must be considered.<br>- Reducing fuel on public and private lands, as well as in the home ignition zone, must considered. |
| 2 | Explore the sensitivity of the exposed community [84]. | - The population of the area, the gender, age, and type of residents must be known.<br>- Their differentiated vulnerabilities need to be considered.<br>- Risk factors of population, such as poverty, age, education, language and special needs, must be known. |
| 3 | Consider the levels of the adaptive capacity [84]. | - Resources available to respond and recover from a wildfire must be provided.<br>- Population available resources to respond and recover from a wildfire must be published. |

**Table 1.** *Cont.*

| No | Principle | Measures |
|---|---|---|
| 4 | Tailored fire prevention and emergency warning [48]. | • The fire prevention and emergency warning must be tailored to specific groups. |
| 5 | Real-time mass communication [48] | • Fire-prevention strategies should include public education and real-time mass communication |
| 6 | Devolution of political power [48] | • Communities need advice on managing fire risk. This requires devolution of political power from centralized bureaucracies to local organizations. |
| 7 | Think diversity [84] | • Promotion of disciplinary, sectoral, social and gender diversity among fire scientists, policymakers and wildfire managers must established |
| 8 | Increase participation [85]. | • The fire prevention and emergency warming plans need to plan with the engagement with local communities and inclusion of women. |
| 9 | Boost gender equality [86]. | • Climate change actions need to be inclusive, based on consultation, participation and knowledge sharing with women to provide opportunities for improving health, education, and livelihoods.<br>• Increasing women's participation would result in more environmental and productivity gains with mutual returns across the SDGs, including SDG 5 and SDG 13. |
| 10 | Include public education [48] | • Fire-prevention strategies should include public education and real-time mass communication |
| 11 | Post disaster Waste management | • Careful waste recycling should be implemented.<br>• Frequent collection of wastes must be implemented.<br>• Post-fire wastes need appropriate and diverse management options. |

### 6.2. Collaboration and Participation

To address wildfire vulnerability in the community and boost resilience, collaboration and participation are needed. Recommendations for an effective collaboration are provided in Table 2.

**Table 2.** Recommendations for an effective collaboration.

| No | Recommendations |
|---|---|
| 1 | Co-work with residents to reduce flammability in their community. |
| 2 | Co-share information with residents. |
| 3 | Collaborate with them to develop trust. |
| 4 | Learn from them about their special needs. |
| 5 | Co-explore the barriers of economic, gender, age, and language, within the community. |
| 6 | Co-work with owners to reduce the risk of their home burning, increase defensible space around their homes. |

**Table 2.** *Cont.*

| No | Recommendations |
|:---:|:---|
| 7 | Collaborate with the waste management planners and local/regional administration on how to optimize the collection and recycling of wastes, based on these assessments. |
| 8 | Collaborate with forestry administration on how to clean forest from flammable bio residues and create a partnership with a local utility company to remove dead trees near power infrastructure. |
| 9 | Identify pre-attack zones for evacuation planning and develop clear maps showing water supply and equipment staging areas. |
| 10 | Collaborate with the administration of urban planning and constructions to define the standards and type of building material for the area in risk. |
| 11 | Create a partnership with a local utility company to remove dead trees near power infrastructure, reduce roadside fuels along strategic private pathways. |
| 12 | Keep pathways free and clear with clear guiding signs. Wider driveways to help residents to evacuate faster. |
| 13 | Improve evacuation preparedness based on these assessments. |

### 6.3. Gender-Sensitive Approaches

Finally, a change leap towards gender-sensitive approaches should be taken by integrating the following in the resilience plan (Table 3):

**Table 3.** Recommendations for a change towards taking gender-sensitive approaches.

| No | Gender-Sensitive Approaches |
|:---:|:---|
| 1 | Employ women at all levels of fire management, waste, and forestry management. |
| 2 | Build capacity for women of the area to acquire the skills and knowledge for fire management. |
| 3 | Develop roles for women to act appropriately in a hazard and post-hazard time. |
| 4 | Co-create respect and trust in women's ability. |
| 5 | Empower women to stand up for their knowledge and experience. |
| 6 | Create a more diverse view of resilience by using other social unequal relationship to fire. |
| 7 | Collaborate to uplift women to participate in fire management and planning by changing the historical, cultural, or political bias behind them. |
| 8 | Vote for those that express a strong political will for climate resilience societies, could be regarding the complex issues of climate adaptation. |
| 9 | Keep all stakeholders closely involved. |
| 10 | Train the community to understand the mitigation plan and recognize the danger of wildfire, and to recognize women's power. |
| 11 | Enhance the quality and scope of team discussions, by keeping them open, equal, and not dominated by ideologies. |

### 6.4. Positive Arguments

Some positive arguments for the discourse of gender equality are depicted in Table 4.

**Table 4.** Positive arguments for the discourse of gender equality in wildfire's resilience.

| No | Positive Arguments |
|----|--------------------|
| 1 | Women fire scientists have contributed significant new insights and perspectives for the future of fire science. |
| 2 | Women have the knowledge and understanding of what it takes to adapt to changing environmental conditions in order to identify practical solutions. |
| 3 | Women bring more empathy and inclusion in defense and problem solving, which enhances their effectiveness as sustainability leaders. |
| 4 | Women remain largely untapped due to existing prejudices. |
| 5 | To effectively mitigate climate change, we need to harness the knowledge and skills of women. |
| 6 | Women have the right to equal participation. |
| 7 | Once in leadership roles, they can make a difference that benefits whole societies. |

## 7. Conclusions

The impacts of gender inequality, women's socioeconomic disadvantages, and the role of socio-cultural norms can limit women from acquiring the information and skills necessary to escape or avoid hazards, as they continue to be ignored. These remain a critical challenge to climate change impacts adaptation efforts in Greece. They render women disproportionately vulnerable to disasters and climate-change hazards.

This study provides a discourse for integrating gender equality dimensions into wildfire disasters planning and management for making the world more resilient to climate changes. It is important that mitigation and adaptation efforts integrate gender issues at all levels.

The study is based on a place-specific instrumental case study (the wildfire of 2018 in Attika, Greece) and provides insights that can help raise awareness on complex and dynamic interactions, the meanings of social–ecological resilience attached to it, and the gender equality dimension in fire organizations in Greece and Mediterranean countries. However, although place-specific and context-specific, this study can be useful at a global scale.

Every crisis includes an opportunity for change. Wildfires of 2018 in Greece should be seen as an opportunity for organizations and communities in the country to learn from and take actions to limit their vulnerability in the future. It is also an opportunity to accelerate diversity and inclusion at all levels, from community preparedness to leadership and take gender-sensitive approaches at all levels.

For developing more resilient communities, planning is needed based on the lessons learnt from this place-based case study and from global experiences. Actors contributing to wildfire vulnerability are both social and ecological and they need strategies for resilience planning, with consideration of some equitable principles that were provided in this study as recommendations.

**Author Contributions:** All authors contributed to several aspects of the study, specifically, conceptualization, A.Z., A.P., and E.-A.K.; methodology, A.Z.; formal analysis, investigation, resources, and data curation A.P., E.-A.K., and A.Z. writing—original draft preparation A.Z.; writing, review and editing, A.Z.; supervision A.Z. All authors have read and agreed to the published version of the manuscript.

**Funding:** Funding received from the European Union's Horizon 2020 research and innovation program under grant agreement No. 741,672 (TARGET project), where RMEI is a partner.

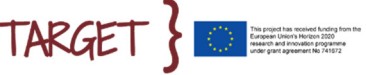

**Institutional Review Board Statement:** Not applicable.

**Informed Consent Statement:** Informed consent was obtained from all subjects involved in the study.

**Data Availability Statement:** Data sharing not applicable.

**Acknowledgments:** Sincere thanks go to the interviewed participants for their time and invaluable contributions. Massimo Guarascio leader of GAMe-RMEI and professor at Sapienza University of Rome and Monica Cardililli are acknowledged for the organization of the MICHELANGELO2019 Workshop—Natural Hazards Risks, Resilience, and Gender Inequalities-Building Bridges within the Mediterranean and TARGET project' in Rome, in 2019. All members of GAMe (Young Mediterranean Ambassadors) and RMEI members (Network of the Mediterranean Engineering Schools) are acknowledged.

**Conflicts of Interest:** The authors declare no conflict of interest.

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
