# Peer review of "Observational Evidence of the Need for Gender-Sensitive Approaches to Wildfires Locally and Globally: Case Study of 2018 Wildfire in Mati, Greece"

_sustainability, doi:10.3390/su13031556_

Round 1
Reviewer 1 Report
I do not have any further comments. The authors have addressed all my prior comments.
Author Response
Dear reviewer,
Since there are no more comments from you, I accept all the corrections performed and I would like to thank you for your time and effort to review and accept our manuscript.
Reviewer 2 Report
Overall, I thought this was an interesting and useful paper. I agree that more attention should be paid to gender diversity and equity when it comes to risk perception and natural hazards, especially climate change. I just have a couple of notes:
It would be useful to at least mention research on gender differences in climate change and risk perception. Here are a few citations:
McCright, A. M. (2010). The effects of gender on climate change knowledge and concern in the American public. Population and Environment, 32(1), 66-87.
Knight, K. W. (2019). Explaining cross-national variation in the climate change concern gender gap: A research note. The Social Science Journal, 56(4), 627-639.
Gustafson, P. E. (1998). Gender Differences in risk perception: Theoretical and methodological perspectives. Risk analysis, 18(6), 805-811.
Also, the choice to use the term "wildfire2018" or "Mati2018" wildfire seems odd to me. It would improve clarity and readability if you referred to the specific wildfire on which the case study was conducted as "the 2018 wildfire" or "the 2018 Mati wildfire" or something similar.
Finally, the paper could use a thorough proofreading for minor typos and grammatical errors. I should note that it was somewhat difficult to read the paper with all of the tracked changes left in. In the future you should submit clean versions of papers for review.
Author Response
Dear reviewer,
Since there are no more comments from you, I accept all the corrections performed and I would like to thank you for your time and effort to review and accept our manuscript.
This manuscript is a resubmission of an earlier submission. The following is a list of the peer review reports and author responses from that submission.
Round 1
Reviewer 1 Report
Please see attachment.

Reviewer 2 Report
There are major problems with English language use throughout the paper that confuse the points and the argument .
The paper lacks focus and rambles through various related topics without tying an argument together or even indicating a clear direction. What is the objective/research question?
The introduction introduces concepts without defining them and fails to set out the structure of the paper.
The discussion of conceptual frameworks is rambling and disjointed and much too long. It fails to provide a useful, coherent framework.
The research methodology section is under described. The method is totally inadequate. Three interviews would be inadequate in most cases but the paucity of results reported from the data collection makes a farce of the section on discussion and lessons learnt. There are almost no data supporting the recommendations and conclusions.
Reviewer 3 Report
The topic and approach of the authors’ research is interesting and compelling. Conceptualizing wildfire resilience within a context of gender equality is important for future management decision making. This submitted research contribution also complements the larger movement to promote gender-responsive climate policy, but from a focused natural disaster perspective. There are some grammatical issues throughout the manuscript, although it reads well overall. These issues must be addressed along with structural issues that begin in section 2.3 with disjointed paragraphs and a reliance on bulleted points in multiple sections. Currently, these issues distract from the focus of the paper. While the idea of the paper is potentially a critical contribution, the execution of the paper could use additional work. More interviews are needed before generalizing trends and perspectives. This would strengthen the argument considerably. The recommendations offered for resilience planning could be extremely valuable if adopted, which supports the importance of this type of research.
Pg1 The abstract has some grammatical issues. For example, in the second sentence: “It aims contributing to reduce…”. The manuscript must be checked for writing errors throughout, but it overall reads well.
Pg6 Line260 Several paragraphs starting with this section (2.3, 2.6, etc.) are very short with some even being one sentence in length. This needs to be rethought and organized for improvement.
Pg10 Line 404 Concern that the detail in listing who was interviewed would reveal their actual identity since it is very specific.